# Serum Metabolites Responding in a Dose-Dependent Manner to the Intake of a High-Fat Meal in Normal Weight Healthy Men Are Associated with Obesity

**DOI:** 10.3390/metabo11060392

**Published:** 2021-06-16

**Authors:** Ueli Bütikofer, David Burnand, Reto Portmann, Carola Blaser, Flurina Schwander, Katrin A. Kopf-Bolanz, Kurt Laederach, René Badertscher, Barbara Walther, Guy Vergères

**Affiliations:** 1Food Microbial Systems Research Division, Agroscope, Schwarzenburgstrasse 161, 3003 Berne, Switzerland; david.burnand19@gmail.com (D.B.); reto.portmann@agroscope.admin.ch (R.P.); carola.blaser@agroscope.admin.ch (C.B.); flurina.schwander@gmail.com (F.S.); katrin.kopf@bfh.ch (K.A.K.-B.); rene.badertscher@agroscope.admin.ch (R.B.); barbara.walther@agroscope.admin.ch (B.W.); guy.vergeres@agroscope.admin.ch (G.V.); 2Instrumat AG, Ch. de la Rueyre 116-118, 1020 Renens, Switzerland; 3Nisco ApS, 2630 Taastrup, Denmark; 4School of Agricultural, Forest and Food Sciences HAFL, Bern University, 3052 Zollikofen, Switzerland; 5Department of Visceral Surgery and Medicine, University Hospital of Berne, 3010 Berne, Switzerland; kurt.laederach@dbmr.unibe.ch; 6Halteneggweg 3, 3145 Niederscherli, Switzerland

**Keywords:** obesity, postprandial challenge, caloric-dose response, serum metabolome

## Abstract

Although the composition of the human blood metabolome is influenced both by the health status of the organism and its dietary behavior, the interaction between these two factors has been poorly characterized. This study makes use of a previously published randomized controlled crossover acute intervention to investigate whether the blood metabolome of 15 healthy normal weight (NW) and 17 obese (OB) men having ingested three doses (500, 1000, 1500 kcal) of a high-fat (HF) meal can be used to identify metabolites differentiating these two groups. Among the 1024 features showing a postprandial response, measured between 0 h and 6 h, in the NW group, 135 were dose-dependent. Among these 135 features, 52 had fasting values that were significantly different between NW and OB men, and, strikingly, they were all significantly higher in OB men. A subset of the 52 features was identified as amino acids (e.g., branched-chain amino acids) and amino acid derivatives. As the fasting concentration of most of these metabolites has already been associated with metabolic dysfunction, we propose that challenging normal weight healthy subjects with increasing caloric doses of test meals might allow for the identification of new fasting markers associated with obesity.

## 1. Introduction

Evaluating the postprandial response to food or meal intake is a key strategy to identify and characterize intake [1,2,3] and effect [4,5,6] as well as susceptibility biomarkers [7,8,9] in nutritional studies [10]. In particular, among the key tools available to validate these biomarkers, challenge studies provide insights into temporal effects, allowing a better evaluation of a causal relationship between dietary intake and metabolic response [11]. Surprisingly, however, the relationship between the dose of the ingested foods or meals and the postprandial response of the organism, a key criterion in biomarker validation, has been little investigated in acute nutrition studies [12]. Indeed, statistical power analyses calculating the number of subjects to be enrolled in acute intervention studies to demonstrate significant changes in the concentration of selected biomarkers in the participants in response to nutritional stimuli generally do not take the dose of the ingested foods or meals into account [12]. This gap should thus be filled in order to obtain better insights into nutritional interactions, especially since linearity in the dose response is not necessarily expected in light of the broad chemical composition of foods and meals and complexity of human physiology.

In order to address this question, we have previously reported a controlled, randomized, crossover acute intervention study in normal weight (NW) and obese (OB) men, having consumed three caloric doses (500, 1000, 1500 kcal) of a high-fat (HF) meal [13]. The postprandial response of these subjects was analyzed within the first 6 h post-consumption by measuring a series of 10 clinical blood markers of metabolic, inflammatory, and hormonal processes. This study did not reveal a common sensitivity of the clinical markers to the caloric increase in the HF meal or to the metabolic status of the subjects, as the ability of these markers to differentiate among the various conditions differed for each of them. Thus, this study indicated that the integration of a dose-response strategy into the design of acute challenge studies might lead to the revisions of conclusions derived from interventions usually conducted with single doses. It also indicated that the dose response of the human organism to foods and meals cannot be globally understood by analyzing a small subset of clinical markers.

Due to key technological developments in mass spectrometry and chemometrics, the field of nutrimetabolomics now provides nutrition scientists with powerful tools to more holistically investigate the impact of nutrition on the human organism [14,15] and its metabolic dysfunctions, such as type 2 diabetes [16], metabolic syndrome [17], and obesity [18,19]. The postprandial response of clinical chemistry parameters in serum [13] and genes in blood cells [20] in NW and OB men to increasing caloric doses of an HF meal has been previously reported. The current report complements these analyses with an untargeted analysis of the serum metabolome of 15 NW men (referred to hereafter as the NW group) and 17 OB men (referred to hereafter as the OB group) (Figure 1). The aims of this analysis are to (i) investigate overall the dose-dependency of the postprandial response of healthy men to a HF meal, (ii) compare the postprandial response of subjects differing in their metabolic health status, namely the NW and OB men, and (iii) investigate whether the postprandial serum metabolome can be used to identify markers that discriminate NW and OB men under fasting conditions. This last aim is motivated by the hypothesis that metabolites increasing their iAUC in response to increasing caloric doses of the HF meal challenge could potentially accumulate over time, in particular in subjects with metabolic disorders, such as obesity, which are characterized by a loss in metabolic flexibility [21,22,23].

## 2. Results

From deconvolution with Progenesis, a total of 29,743 features could be integrated and evaluated. Only features present in more than 50% of the QC samples were used for statistics. These 25,877 features had to pass additional filtering steps; that is, the coefficient of variation of features in the QCs had to be below 30%, and the mean value of the QC had to be at least three times more than the average blank injection. After these filtering steps, 2235 features remained in the active dataset.

A PCA analysis of the 2235 features measured in all serum samples identified five outlying samples from four subjects in the NW group (subject 12: meal A; subject 14: meal C; subject 16: meals A and C; subject 19: meal A). As the statistical workflow relied on a comparison of the iAUC across all three doses of the HF meal, the complete datasets of these four subjects were removed from further analysis. Among the 2235 features, 1024 demonstrated a significant postprandial response after at least one dose of the HF meal in the NW group. Finally, among the 1024 postprandial features, 135 demonstrated a significant difference in the iAUC between at least two doses of the HF meal in the NW group. Details of the fasting and postprandial behavior of the 135 dose-dependent features in the NW and OB groups as well as of their mass signal (*m/z*) are shown in Appendix A.

The kinetic profile of the 135 dose-dependent features measured in the NW group is shown, together with their profile in the OB group in Figure 2**.** Most of the features demonstrated a positive (iAUC > 0) postprandial effect (107 from 135 features). Globally, the kinetics patterns of these molecules were separated into four clusters, clusters 1–3 being characterized by postprandial increase, each with a different kinetic profile, and cluster 4 by a decrease. In cluster 1, many of the features showed a strong dose-response effect for the NW and OB groups. Cluster 2 was characterized by a significant postprandial increase up to 4 h at all three doses and by a slight decrease after 6 h for the NW group, the postprandial effects being weaker in the OB group. Features in cluster 3 were characterized by a late increase in their concentration, with a maximum at 4 to 6 h. Features with negative postprandial responses were located in cluster 4. At first sight, a qualitative global analysis of the kinetic profiles revealed a reproducible postprandial response of the 135 features irrespective of the ingested dose of the HF meal or investigated group (i.e., NW or OB group). On the other hand, a closer visual inspection of the profiles revealed numerous differences due to these conditions. For example, the postprandial response of the features in cluster 1 after ingestion of 1000 and 1500 kcal HF meals appeared higher than after 500 kcal in both the NW and OB groups. In addition, the features in cluster 2 appeared higher under fasting conditions in the OB group than in the NW group. In addition, the concentration of these features appeared to return to baseline 6 h after ingestion of 500 and 1000 kcal HF meals in the NW and OB groups. Most features in cluster 2 did not return to baseline concentrations until 6 h after intake of the 1500 kcal HF meal in the NW and OB groups.

Given the large number of sampling conditions in this study (five time points, three caloric doses, and two groups of subjects), a detailed quantitative analysis of the differences illustrated by clusters one and two presented in the previous paragraph was not pursued. Rather, the dataset was simplified by conducting a statistical analysis of the overall postprandial response of the 135 features, that is, by characterizing their iAUC. A cluster of the iAUC of the 135 dose-dependent features measured in the NW group is shown, together with the corresponding iAUC in the OB group, in Figure 3. The right panel of Figure 3 also illustrates results from a statistical evaluation of these iAUCs. In agreement with the results of Figure 2, the iAUC of the 135 features were grouped in two clusters, the major cluster 1 composed of features with a postprandial increase, and the minor cluster 2 composed of decreasing features. Notably, the statistical analysis of the NW group showed that most of the significant changes in iAUC were observed when comparing the response after 500 kcal of the HF meal to the response after 1000 or 1500 kcal, but no longer when comparing the responses after 1000 to 1500 kcal. The OB group followed a similar, although less pronounced, pattern. Furthermore, the switch from 500 to 1000 kcal in the NW group was accompanied by an increase in the iAUC of the features in cluster 1 but by a decrease in the features of cluster 2. Here also a similar, less pronounced, pattern was observed in the OB group. In line with this last observation, a statistical comparison of the iAUC of the NW and OB groups for each of the ingested doses of the HF meal showed that almost all the iAUC were smaller in the OB group compared with the NW group. This phenomenon was most evident in the comparison of the iAUC after the ingestion of the 1000 kcal HF meal, with 63 out of 135 features demonstrating significantly (*p* < 0.05) lower iAUC in the OB group than in the NW group, whereas none demonstrated significantly higher iAUC.

To investigate whether the postprandial serum metabolome of NW men can be used to identify markers that discriminate NW and OB men under fasting conditions, we next compared the fasting values of the 135 dose-dependent features in the NW and OB groups. Among the 135 dose-dependent features measured in the NW group, 52 features were significantly different (Kruskal–Wallis rank sum test, *p* < 0.05) at fasting (t = 0 h) between the NW and OB groups (Appendix A, Figure 3). Strikingly, all 52 features were higher in the OB group, with the concentration of the features in the NW group ranging from 57 to 96% of the fasting concentration in the OB group. Moreover, a correlation analysis between the fasting concentration of the 52 features and their iAUC after ingestion of the three doses of the HF meal in the NW and OB groups (analyzed together) revealed an almost exclusively significant negative association between the fasting and postprandial behavior of a majority of them. This association remained for a significant number of features when the NW and OB groups were investigated separately and for each dose of the HF meal, as shown by the NW group that ingested the 500 kcal HF meal. Details of these correlations are shown in Appendix A.

Among the 52 features, eight could be identified at level 1. Four of them were proteogenic amino acids (alanine, arginine, lysine, and valine), and one was a non-proteogenic amino acid (homoarginine). A compound identified at level 3 as threonine, or its isomer homoserine, was also detected. The remaining two compounds identified at level 1 were the amino acid derivative creatine and allantoin, a diureide of glyoxylic acid. The postprandial kinetics of each of these eight compounds are shown in Figure 4**.** Eight additional proteogenic amino acids (cystine, isoleucine, proline, threonine, phenylalanine, histidine, glutamic acid, and tyrosine) and two non-proteogenic amino acids (beta-alanine and ornithine) measurable by GC-MS were evaluated to extend the analysis of amino acids beyond the ones described above (Appendix A).

With the exception of threonine, the fasting concentration of these amino acids was elevated in the OB group compared with the NW group, although only isoleucine, tyrosine, and glutamic acid were significantly higher in the OB group. In addition, in agreement with the results of the amino acids measured by LC-MS, the iAUC of most of the amino acids measured by GC-MS (beta-alanine, isoleucine, threonine, phenylalanine, ornithine, histidine, tyrosine, and glutamic acid) were elevated in the NW group compared with the OB group after at least two HF meals.

Correlation analyses were conducted on the postprandial iAUC of the 52 features characterized above and the iAUC of the clinical markers (glucose, insulin, triglycerides, cholesterol, HDL-cholesterol, C-reactive protein (CRP), interleukin-6 (IL-6), endotoxin, and glucagon-like peptide-1 (GLP-1)) reported earlier [13]. Appendix A shows the postprandial characteristics of these clinical markers for the samples analyzed in this report. The correlation analysis showed positive correlations remaining significant after false discovery rate (FDR) correction for multiple testing for most of the 52 features with insulin and triglyceride as well as negative correlations with HDL-cholesterol in the NW group. Details of these correlations are shown in Appendix A. Correlation of the same clinical markers at fasting and the fasting results of the 52 features measured in this report showed only a few significant correlations after FDR correction (data not shown).

## 3. Discussion

The fasting metabolome has become a major tool for differentiating lean or NW from OB subjects as well as healthy subjects from subjects with a dysfunctional metabolism [24]. As metabolic dysfunction is often mechanistically linked to nutrition, the coupling of metabolomics analyses to challenge tests based on the ingestion of nutrients, such as glucose (OGTT) and lipids (OLTT), foods, or meals, provides an additional sensitive way to evaluate the metabolic status of the human organism [25,26] as well as its response to dietary treatment [27], including weight loss programs [28]. This research strategy has fostered the development of personalized nutrition, and concepts such as the “health space” [29,30] and “phenotypic flexibility” [21,22,23] have emerged. In this evolving context, our report is the first metabolomics study that integrates a dose-response component into the nutritional challenge.

Although non-linearity in the dose response has already been demonstrated for individual dietary markers [28,31,32,33,34,35], the panel of dose-dependent features responding to the HF meal in the NW group showed a remarkable global saturation of the postprandial metabolome between 1000 and 1500 kcal. Furthermore, although the overall dose response of the OB group was quantitatively similar to the NW group, the postprandial metabolome of the OB group was globally characterized by a decreased response, particularly after ingestion of the 1000 kcal HF meal. Interestingly, the strategy of selecting dose-dependent postprandial features led to the identification of a subset of 52 features discriminating the NW and OB groups under fasting conditions by all being significantly higher in the OB group compared with the NW group. The predictive potential of this set of 52 features was further suggested by the observation that the fasting and postprandial concentration of these features were almost exclusively negatively correlated not only in the OB group but also in the NW group.

Whereas the NW and OB groups received the same caloric doses, differences in the fasting and postprandial behavior of these two groups could not be explained by blood volume and/or basal metabolic rates, so that other factors, gastrointestinal and/or systemic, could explain the quasi-unidirectional behavior of the 52 features. One could hypothesize that some control mechanisms associated with phenotypic flexibility [21] were at play, which prevented metabolites present at already high fasting concentrations in OB subjects from increasing postprandially as much as in NW subjects. Translating the set of 52 features into useful biomarkers of metabolic dysfunction requires, however, that the functionality and causality of the findings be clarified. A prerequisite for this discussion is, consequently, the molecular identification of these features.

Nonetheless, among the dose-dependent features with increased fasting serum concentration in the OB group, seven amino acids (alanine, arginine, glutamic acid, isoleucine, lysine, tyrosine, valine) were identified. Increased concentrations of fasting amino acids in the blood of OB subjects, both healthy and unhealthy, has been well established, in particular, but not exclusively, for branched-chain amino acids (BCAAs) [19,36,37,38,39,40,41,42,43]. In particular, increased fasting blood concentrations in OB subjects have already been reported for each of the seven amino acids identified in this report, including the BCAAs valine and isoleucine [36,37,38,40], alanine [36,37,40], lysine [44], threonine [37,44], arginine [40], and glutamic acid [37]. Importantly, the postprandial sensitivity of amino acids, including BCAAs, to a dietary challenge also depends on the metabolic status of the subjects, including their body mass index (BMI) [26,27,30,45,46]. The mechanism predominantly proposed to relate changes in BCAAs to metabolic dysfunction is a crosstalk among these amino acids, glucose, and lipid homeostasis, via different mechanisms involving the mammalian target of rapamycin (mTOR) pathway [41,46] and gastrointestinal hormones [33,47] and leading to insulin resistance as a result of mitochondrial overload [48,49]. In this context, the significant correlations observed in our study between the postprandial concentration of a majority of the 52 features and postprandial insulin (positive), triglycerides (positive), and HDL-cholesterol (negative) are in line with associations already reported for subjects with metabolic dysfunction under fasting conditions for insulin [40,50], triglycerides [51,52,53,54,55,56], and high-density lipoprotein (HDL)-cholesterol (65), as well as postprandially [52,56]. Strikingly, these associations were observed in the NW group, but were much weaker, and less significant, in the OB group, in line with the conclusion that OB subjects react to the increasing doses of HF meals with altered efficiency, which supports the already proposed model of metabolic inflexibility and mitochondrial overload resulting from overfeeding [57].

In addition to proteogenic amino acids, amino acid derivatives were identified among the 52 features. The identification of the non-proteogenic homoarginine is interesting as it is derived from two amino acids also present among the 52 features, lysine and arginine, and its association with cardiovascular diseases [58] and obesity [59] is currently debated in the literature. Homoserine has not been associated with obesity, but lower blood concentrations were reported in patients with type 2 diabetes compared with healthy controls [60]. Creatine is an alpha-amino acid derivative, present in animal foods but also synthesized from arginine and lysine, which serves as a substrate for ATP synthesis [61] and has a role in energy expenditure, diet-induced thermogenesis, and defense against diet-induced obesity [62]. Interestingly, a lower concentration of blood creatine was observed in metabolically healthy OB women after a lifestyle intervention for weight loss [63]. The last molecule identified was allantoin, an oxidation product of uric acid and purine metabolism used as a marker of oxidative stress [64]. Evidence for a link between allantoin and obesity is not available. In fact, a study investigating the blood concentration of a panel of biomarkers of oxidative stress in metabolically healthy and unhealthy OB subjects as well as in NW subjects did not reveal a discriminating role for allantoin [65]. On the other hand, high plasma allantoin levels were found to be inversely associated with type 2 diabetes risk [66].

The current study has a number of limitations. First, although the study was sufficiently powered to detect a significant number of features fulfilling the aims of this study, the relatively small sample size calls for the confirmation of the representability of our findings in another cohort. Second, both gender and age are important factors explaining metabolome variability [67]. The recruitment was therefore restricted to men, and the participants in NW and OB groups were age-matched to reduce this variability. Third, four outliers were identified in the NW group and eliminated from the data analysis, and it was not clear whether these outliers were due to analytical issues or indicative of biological differences between the subjects. Fourth, the method used in this report for the GC-MS analysis requests a manual integration of the features of interests so that a quantitative analysis of the untargeted GC-MS metabolome, similar to the one conducted by LC-MS, was not possible. Hence, the focus for the GC-MS data was set on a targeted quantification of amino acids complementing the amino acids already identified by LC-MS. Finally, among the 52 dose-dependent features differentiating the NW and OB groups under fasting conditions, we could only identify a small subset of them. The identification of interesting features represents, however, a major bottleneck, which is shared by the metabolomics research community [14].

Nonetheless, our study presents, for the first time, the response of the human serum metabolome to increasing doses of a meal. The postprandial response of NW subjects to increasing caloric doses of an HF meal allowed the identification of metabolites, in particular amino acids, that are associated with obesity. A negative correlation between the fasting and postprandial concentration of these metabolites in NW subjects suggests that they could be used as early diagnostic tests of metabolic health. Additional efforts to identify further metabolites as well as to validate the behavior of the dose-dependent metabolites in an independent cohort, including women, are, however, necessary. The impact of changing the relative macronutrient composition of the challenge meal on the dose-dependency of the metabolites would also be worth investigating.

## 4. Materials and Methods

### 4.1. Subjects

Nineteen NW men (BMI: 20–25 kg/m^2^) and 18 OB men (BMI: >30 kg/m^2^) from 25 to 55 years of age were recruited from the Berne area of Switzerland (Figure 1). The number of participants per group was determined based on a power analysis to observe a significant postprandial increase in interleukin-6 after intake of the HF meal [13]. Previous reports on acute intervention studies with foods have shown that 11–17 subjects per group are sufficient to detect, by untargeted metabolomics, significant postprandial changes in the concentrations of several hundred, if not thousand, postprandial features, a significant fraction of them differentiating between the ingestion of different foods [1,2,3,68]. In addition to BMI, the OB group at baseline was characterized, compared to the NW group, by significantly higher weight, waist circumference, and visceral adiposity index, as well as higher blood values for glucose, insulin, HOMA-IR, triglycerides, total cholesterol:HDL cholesterol, C-reactive protein (CRP), and glucagon-like peptide 1 (GLP-1). The OB group had lower blood concentrations of HDL cholesterol. Height as well as blood total cholesterol, interleukin-6 (IL-6), and endotoxin were not different. The two groups were age-matched so that the mean age was not significantly different between the NW and OB groups [13]. See the report by Schwander et al. [13] for a tabulated overview of the baseline characteristics of the participants in the study. [1,2,3,13,68]. The subjects were not allowed to take dietary supplements from two weeks prior to the start of the study until its completion [13].

### 4.2. Study Design

The human study was undertaken at the University Hospital, Berne, Switzerland. Approval for the study was obtained from the Ethics Committee of the Canton of Bern (KEK number 006/11). The study was reported according to the checklist published in the Consolidated Standards of Reporting Trials Statement. All participating subjects provided informed consent. Exclusion criteria were past or present cardiovascular disease, diabetes, inflammatory condition, medications influencing inflammatory and lipid markers, current smoking, food allergy, and impaired kidney or liver function. Dietary supplements (2 weeks prior to the start of the study until its end) and blood donations (3 months before the start of the study until its end) were not allowed. In the crossover study design, each subject had to ingest three different caloric amounts of the HF meal (500, 1000, 1500 kcal) with at least one week of a washout phase between the test meals (Figure 5).

The order of HF meal consumption was randomized among subjects. One liter of water was provided during the postprandial period, but no additional food was allowed. Blood was collected from subjects who fasted overnight (10 h) prior to each test meal intake from 8 am to 9 am, as well as 1, 2, 4, and 6 h after the beginning of the test meal ingestion. Serum samples were preserved at −80 °C before being analyzed by untargeted liquid chromatography-mass spectrometry (LC-MS) and complemented by gas chromatography-mass spectrometry (GC-MS). Further details on the study design are shown in the report by Schwander et al. [13].

### 4.3. Meal Composition

The HF meal consisted of plain bread, palm fat, salami, and boiled eggs, all obtained from Swiss supermarkets. The three HF meals had the same macronutrient composition, with 60.5% of the energy coming from fat, 21.3% from carbohydrates, and 18.2% from protein, and they differed only in their energy content, that is, 500, 1000, and 1500 kcal. Two hundred, 400, and 600 mL of water (Vittel) had to be drunk during the consumption of the 500, 1000, and 1500 kcal meal, respectively. See the report by Schwander et al. [13] for a tabulated overview of the composition of the HF meal.

### 4.4. Untargeted Metabolomics with LC-MS

The blood serum samples were frozen at −80 °C until analysis. A Phree filter was used to remove proteins and phospholipids (Phenomenex Inc., Torrance, CA, USA). Untargeted analysis was performed on an ultra-performance liquid chromatography system (UltiMate 3000, ThermoFisher Scientific, Waltham, MA, USA) with a qTOF mass spectrometer (maXis 4G+, Bruker Daltonik GmbH, Bremen, Germany). The samples from each participant were run together, and their injection order was randomized. The injection order of the 37 participants was also randomized. The measurements were all performed in the positive mode in the range 75 to 1500 *m/z*. The stability of the MS signals was tested with a control sample (QC), which was a mixture of all serum samples. Contamination of the system was controlled by the injection of ultrafiltered water. The identification of the components was performed with the National Institute of Standards and Technology database (NIST v14), the Human Metabolome Database [69], the MassBank of North America (MoNA) [70], and Metlin [71]. The mass accuracy was set to 5 ppm. The identity of the components was verified using collision-induced dissociation (CID) with a range from 5 to 70 eV collision energy.

The following levels of identification were used [15]. Level 1: The compound was identified based on a difference in retention time below <20%, and the *m/z* difference below 10 ppm was compared with the chemical reference standard. Level 2: The tentative annotation was performed by spectral similarity with public/commercial databases with a pectral matching with an *m/z* difference below 10 ppm, but the reference substance was not available. Level 3: The spectral similarity (*m/z* difference below 10 ppm) was checked with public/commercial spectral libraries. The compound can only be attributed to a compound class. Level 4: The retention time and *m/z* signal were available, but identification was not possible.

### 4.5. Data Processing and Statistical Analysis of Untargeted LC-MS Data

Retention time correction, peak picking, deconvolution, and data normalization (default automatic sensitivity and without minimum peak width) were performed with Progenesis QI (v.2.3.6198.24128, NonLinear Dynamics Ltd., Newcastle upon Tyne, UK). Due to the large number of samples, several batches had to be measured over a longer period. The Locally Estimated Scatterplot Smoothing (LOESS) method was applied for correction to correct the drift over the whole measuring process [72]. Features with a low frequency < 50% in QC samples and a large relative standard deviation (*SD*) >30% in QC samples were excluded from the evaluation, and features (median values) that were less than three times higher in the QC samples than in the blanks were not evaluated [68].

The incremental area under the curve (iAUC) of the features detected by LC-MS was calculated between 0 and 6 h after the intake of each dose of the HF-meal by each subject using the AUC function in R (linear model, R package MESS ver. 0.5.7) [73]. Principal component analysis (PCA) of the iAUC results was conducted, and outliers (subjects 12, 14, 16, 19) were removed from further analysis based on a tolerance of 95% beyond the Hotelling’s T2 ellipse. Postprandial time effects were analyzed with univariate nonparametric analysis of longitudinal data (nparLD f1.ld.f1) for all features (*p* < 0.05) (R package nparLD ver. 2.1) [74].

Among the features with a significant postprandial effect in the NW group, the iAUC were compared with the Kruskal–Wallis rank sum test (FDR < 0.05) to find the dose-dependent features, i.e., those responding to the caloric increment with a significant change in their postprandial response in the NW group.

Median values of the dose-dependent features were calculated for the NW and OB groups at each time point and caloric dose and standardized to the highest median value of each feature. The postprandial kinetics of the dose-dependent features were grouped and visualized for both the NW and OB groups with a hierarchical cluster analysis. Spearman’s distance measure and Ward linkage were used as parameters (R package amap ver. 0.8–18 and dendextend ver. 1.14.0) [75,76].

Pairwise comparisons with the Conover–Iman test (*p* < 0.05) (R package conover.test ver. 1.1.5) were performed between the caloric doses for each of the dose-dependent features identified by the Kruskal–Wallis rank sum test [77] to characterize the dose-dependency in the NW as well as in the OB groups. For each of the three caloric doses of the HF meal, the postprandial response of each of the dose-dependent features identified in the NW group was also compared by the Kruskal–Wallis rank sum test with the iAUC observed after the same caloric dose in the OB group. Finally, for each of the dose-dependent features identified in the NW group, their fasting values were compared to the fasting values of the OB group by the Kruskal–Wallis rank sum test.

Robust Spearman correlation analyses were performed to identify significant correlations (*p* < 0.05) in the NW and OB groups between the iAUC and the fasting values of the dose-dependent features that showed a significant difference between both groups under fasting conditions (Appendix A). Spearman correlation analyses were also performed to identify significant correlations (*p* < 0.05) in the NW group between the iAUC of the dose-dependent features that showed a significant difference between the NW and OB groups under fasting conditions and the iAUC of the clinical variables from the same samples reported by Schwander et al. [13] (Appendix A).

### 4.6. Measures of Amino Acids by GC-MS

Ten amino acids (cystine, isoleucine, proline, threonine, phenylalanine, histidine, beta-alanine, ornithine, glutamic acid, and tyrosine) were analyzed in untargeted GC-MS datasets measured on 14 participants in the study (7 in the NW group and 7 in the OB group) to complement the panel of metabolites identified by LC-MS. The samples were prepared according to the procedure of the Human Serum Metabolome Consortium [78] and measured on an Agilent 7890B/5977 A GC-MS system, 70 eV, equipped with a DB-5MS column 60 m × 0.250 mm × 0.25 μm (Agilent Technologies, Santa Clara, CA, USA) as detailed in Pimentel et al. [79]. The data for each amino acid was analyzed with Agilent MassHunter (Agilent Technologies, V.B.7.00, V.B.8.00) using U13C6-labeled D-Fructose (Cambridge Isotope Laboratories, Inc., Tewksburg, MA, USA) as an internal standard to correct the peak areas of the amino acids prior to statistical evaluation. The statistical analyses of the amino acids measured by GC-MS were conducted as for the LC-MS data, except that it was not corrected for multiple testing.

### 4.7. Clinical Chemistry

The methods for the analysis and the results of glucose, insulin, triglycerides, total cholesterol, HDL cholesterol, CRP, IL6, endotoxin, and GLP-1 in the fasting and postprandial blood samples of the NW (*n* = 19) and OB (*n* = 17) subjects have been reported earlier by Schwander et al. [13]. Appendix A shows the data, and corresponding statistical analyses for the participants analyzed in this study (15 participants in the NW group, 17 participants in the OB group). The statistical analyses of the clinical chemistry data were conducted as for the LC-MS data except that it was not corrected for multiple testing.

## Figures and Tables

**Figure 1 metabolites-11-00392-f001:**
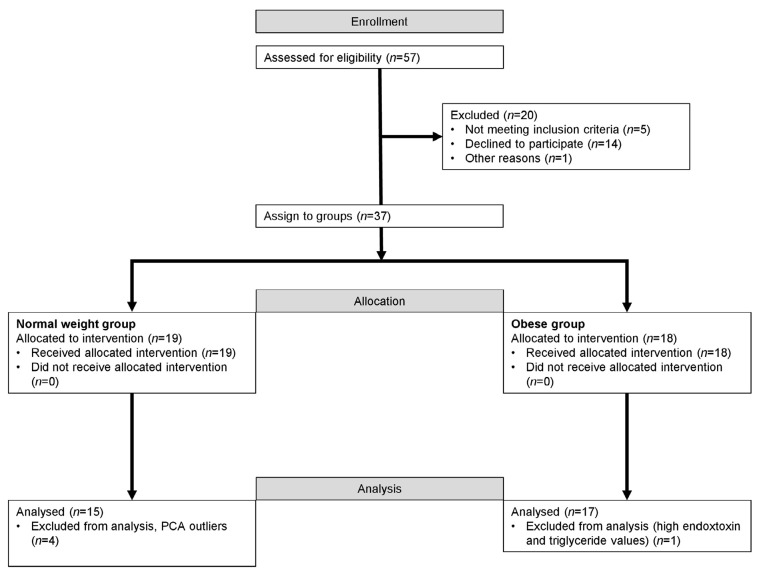
Participating flow chart.

**Figure 2 metabolites-11-00392-f002:**
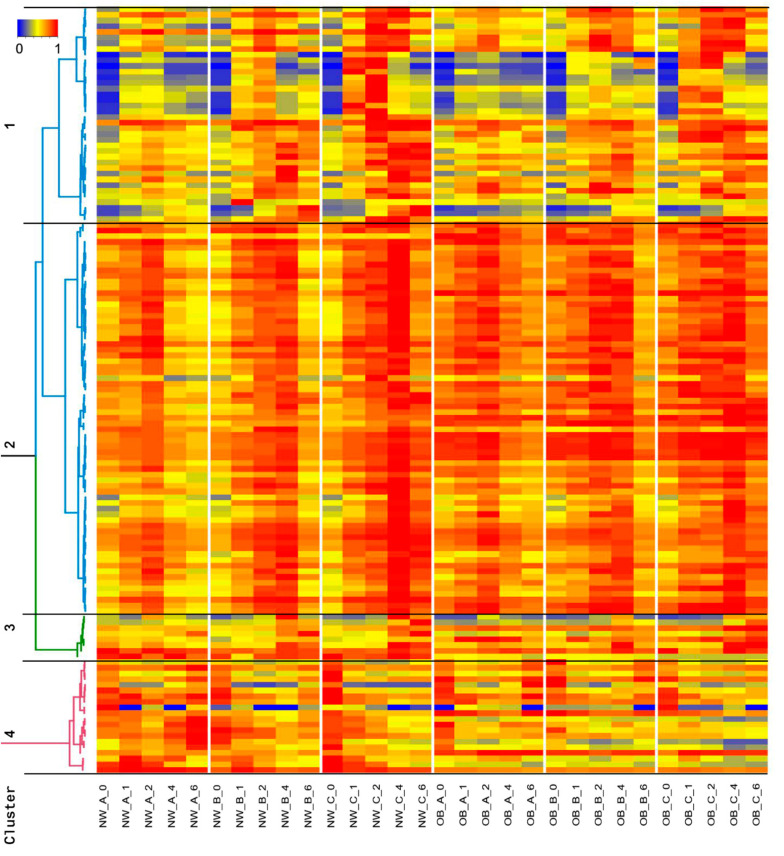
Clustering of postprandial kinetics (0, 1, 2, 4, 6 h) of 135 dose-dependent features in 15 normal (NW) and 17 obese (OB) men after consumption of three caloric doses of HF meals (A = 500, B = 1000, C = 1500 kcal). Median values for all groups and each feature are shown.

**Figure 3 metabolites-11-00392-f003:**
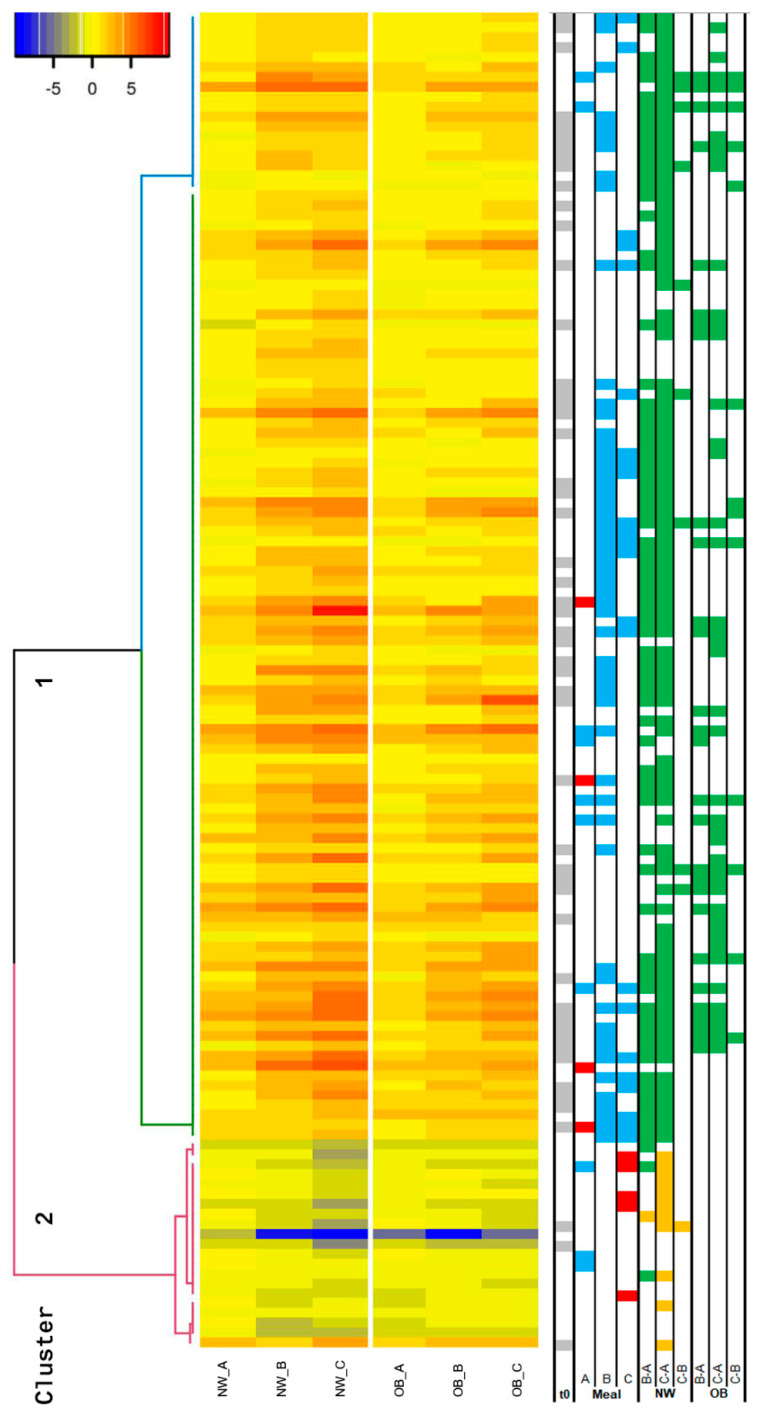
Postprandial iAUC results (0–6 h) of 135 dose-dependent features in 15 normal weight (NW) and 17 obese (OB) men after consumption of three HF meals (A = 500, B = 1000, C = 1500 kcal). Median values for all groups and each feature are shown. Significant (*p* < 0.05) positive ■ difference at fasting (t0) between OB and NW groups (Kruskal–Wallis rank sum test); Significant (*p* < 0.05) positive ■ or negative ■ difference in iAUC between NW and OB groups for meal A, B, or C (Kruskal–Wallis rank sum test); Significant (*p* < 0.05) positive ■ or negative ■ difference in iAUC between different caloric doses B-A, C-A, or C-B for NW and OB groups (Kruskal–Wallis rank sum test).

**Figure 4 metabolites-11-00392-f004:**
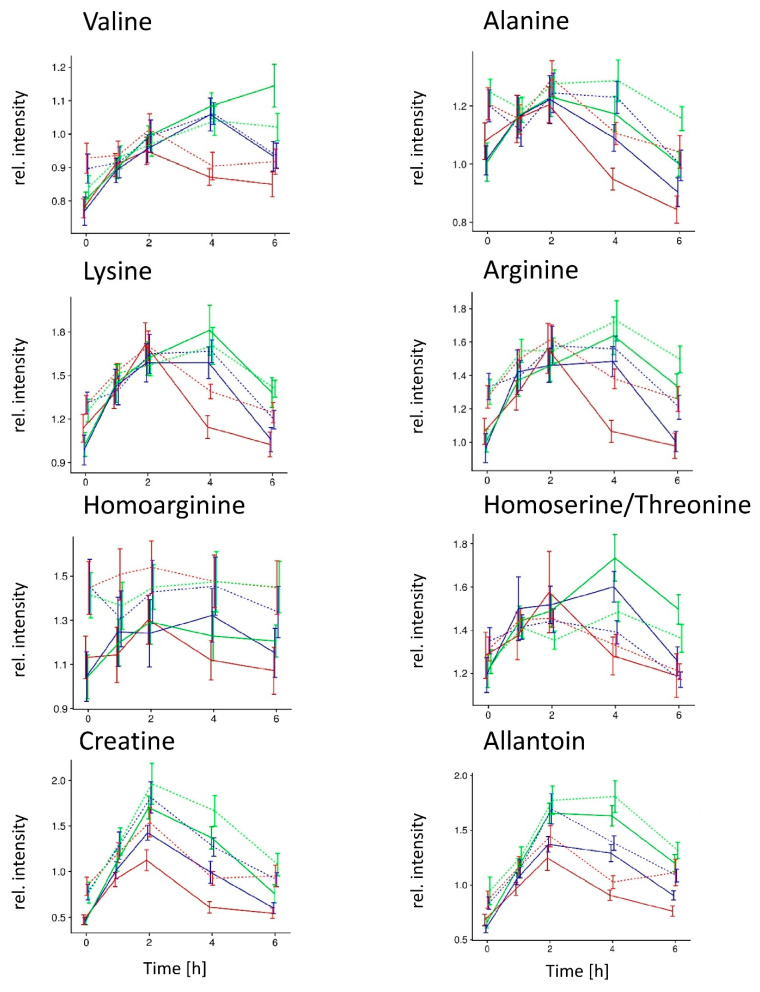
Postprandial kinetics of seven compounds with level 1 and homoserine/threonine with level 3 identification (0–6 h) in normal weight (NW) and obese (OB) group after consumption of three caloric doses of HF meals (A = 500, B = 1000, C = 1500 kcal). The kinetic profile of allantoin after ingestion of 500 kcal of the HF meal by the NW group was dominated at 2 h by an outlier value of a single subject (intensity of 32.6), which was much higher than for all other subjects (intensities ranging from 0.5 to 1.8). This data point was kept for the non-parametric statistical analyses but removed for the graphical representation of the postprandial kinetics of allantoin. 

 NW group; 

 OB group; meals 

 A, 

 B, and 

 C.

**Figure 5 metabolites-11-00392-f005:**
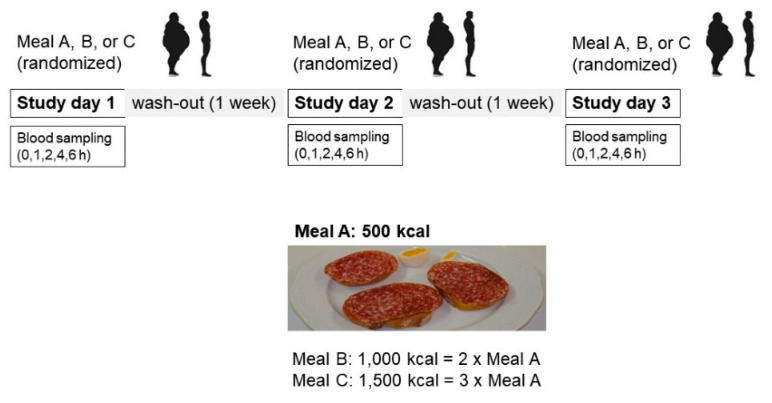
Design of the crossover study.

## Data Availability

The data presented in this study are available on request from the corresponding author.

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
