# Peer review of "Serum Metabolites Responding in a Dose-Dependent Manner to the Intake of a High-Fat Meal in Normal Weight Healthy Men Are Associated with Obesity"

_metabolites, 2021, doi:10.3390/metabo11060392_

Round 1

Reviewer 1 Report

In this manuscript the authors reanalyze data/samples collected as part of the study reported in their 2014 Journal of Nutrition paper to examine serum metabolites that respond to their HF dosage diet paradigm in normal weight and obese patients.  Overall, the data presented in this manuscript are sound and provide a useful extension of insights from their original study.

Minor criticisms:

  1. The overarching argument of the original study and this manuscript is the dose dependence of the metabolites to the dosage of fat provided in the diets. Unfortunately, the way figures 204 are presented, it’s actually quite difficult to see this dose response. I would suggest that:
    • Figure 2 be rearranged so that samples from the different fat dosages are adjacent and the different times points are separated (Similar arrangement to figure 3)
    • In Figures 2 & 3 the color scale (yellow to red) effectively reduces the number of levels that can be discriminated and effectively compresses difference between samples. Changing to a different color scale would make seeing differences associated with the dose response easier.  Also including a scale bar (rather than just individual color boxes in the figure legend) would be helpful.
    • Higher resolution versions of the figures (especially FIgure4) would make it easier to interpret and understand the graphs.
  2. In the Supplemental Table S1, median values are presented.
    • Why was median chosen instead of some mean (arithmetic, geometric, )?
    • Some measure of variance is essential to understanding the date (ideally standard deviation).
    • For the T0 values, a p-value is reported. One assumes this p-value is adjusted for the number of tests (features) measured. If not, it clearly should be.  Either way, a description of this adjustment should be included.
  3. Further explanation of sample and feature exclusion (lines 84-89) would be helpful. Specifically, how was the decision to exclude samples with >30% CV arrived at? Was this a priori decision or was it done after examining the data? Also, how was the decision to excludes samples with less that 3x the blank arrived at?  Was this the limit of quantification or based on some other factor?  This seems important as this secondary QC filtering eliminated more than 90% of the detected features, which certainly has the potential to introduce bias.

Mention of levels of identification (lines 168-171) and legend of Figure 4 is confusing. It is described in the materials and methods section, but it would be helpful to include at least a brief explanation in the results text.

Reviewer 2 Report

The article is an interesting research, well and logically presented, well-chosen illustrative material. Questions to the Authors: 1. Why are only men included in the study? 2. The difference in age is very large (range 25-55 years). Have you checked how the indicators change depending on the age of the volunteers? 3. It is interesting to see how the metabolic profile changes if the protein / fat / carbohydrate ratio in the dietary load is changed. 
